# Moderating Effects of Emotional Recognition Competency in Rejective Parenting and Adolescent Depression and Aggression

**DOI:** 10.3390/ijerph20186775

**Published:** 2023-09-18

**Authors:** Jaeeun Shin, Sung Man Bae

**Affiliations:** 1Department of Psychology, Chung-Ang University, Seoul 06974, Republic of Korea; rheai@cau.ac.kr; 2Department of Psychology and Psychotherapy, Dankook University, Cheonan 31116, Republic of Korea

**Keywords:** rejective parenting, depression, aggression, emotional recognition, RMET

## Abstract

Rejective parenting is a major antecedent of emotional instability and aggressive behavioral problems. Previous studies have reported that emotional problems, such as depression, anxiety, aggression, and conduct problems in children and adolescents, improve through interventions that enhance emotional recognition competency. In this study, we explored whether the emotional recognition ability levels of individual adolescents moderated the pathway of negative parenting on aggression mediated by depression The moderated mediating effect of emotional recognition competency was investigated through examining 2265 first-year high school students using the 2021 data from the Korean Children and Youth Panel Survey 2018. There was no significant moderating effect on the direct pathway from rejective parenting to aggression. The moderating effect of emotional recognition competency on the indirect pathway leading to aggression through depression in rejective parenting was significant. These results suggest that the ability to correctly understand signals related to the emotions of others can play an important role in reducing depression and aggressive behavior by reducing conflict with people around them and experiencing more support.

## 1. Introduction

Aggression during adolescence is one of the most common behavioral problems. High levels of aggression are associated with various psychosocial maladjustments due to the high risk of undermining internal stability and external support networks. In particular, adolescent aggression tends to correlate with depression, and depression and conduct problems are mutually dependent and often interact [1,2]. The acting-out model emphasizes that adolescents’ internalization problems are antecedent variables of externalization problems and that depression, an internalization problem, can be expressed as aggressive behavior. Adolescent depression is often referred to as “masked depression”. Masked depression is defined as an inability to externally communicate one’s thoughts or feelings despite an increased inner sense of depression and expressing one’s mental distress to the outside world through aggressive, externalized behavior [3]. Depression and aggression can also be said to be reciprocal in that externalizing problems such as aggression can further increase depression and cause conflicts with significant others, rejection, and the withdrawal of support. The integrated model suggests that depression and aggressive behavior interact and reinforce one another; however, efforts should be made to identify other potential variables that affect these factors [4].

Previous studies have emphasized the quality of parent–adolescent relationships as one of the obvious potential variables commonly affecting adolescents’ internalization and externalization problems. Perceived parental rejection is defined as an adolescent’s perception that their parents do not care for them. Hale et al. [5] explored the relationship between perceived parental rejection, depression, and aggression and suggested that parental rejection explained adolescent aggression and withdrawal through depression. In addition, meta-analyses have found that perceived parental rejection is associated with overall maladaptive personality traits such as hostility/aggression, dependence, emotional unresponsiveness, emotional instability, and negative worldviews in children across cultures [6].

Parents’ positive influence on their children is considered a protective factor against mental health disorders [7]. However, children who experience rejection from their parents are particularly prone to increased emotional instability, resulting in frequent mood swings and anxiety, depression, and irritability, potentially leading to aggressive behavior problems [8,9,10]. In addition, the aggressive behavior of children, when triggered by negative parenting, can mutually interact with the behavior of parents and, conversely, foster parental criticism and rejection [11]. In addition, in schools, where most of the time is spent during adolescence, maladjustments, such as rejection from peers and interpersonal anxiety, can be experienced [12,13,14]. Parenting attitudes significantly impact children’s emotional instability and resulting mental health problems.

The impairment of emotional recognition is a transdiagnostic index of early mental health problems [15]. On the one hand, previous studies have reported that environmental risk factors such as negative parenting predict poor facial expression recognition processing, which is a representative emotional recognition ability, and that children and adolescents who have experienced attachment trauma or exhibit behavioral problems experience difficulties in recognizing facial expressions [16]. On the other hand, it has been suggested that a child’s emotional recognition competency is an area that can be developed, and when emotional recognition is improved, it has the effect of reducing negative moods such as depression and anxiety, which are considered internalized problems [17]. In another study, emotional and behavioral problems improved in children with high levels of behavioral problems when emotional recognition training (e.g., MindReading [18]) was provided [15]. A study of Korean adolescents (second- and third-grade students in middle school) also revealed that the ability to read the mind moderates the relationship between distress and problems caused by interpersonal trauma. It suggests that the higher the distress caused by interpersonal trauma experiences, the higher the level of conduct problems; however, the higher the ability to read the mind, the lower the level of conduct problems [19]. This is also consistent with previous studies that have shown that the Reading the Mind in the Eyes Test (RMET) is effective in predicting behavioral problems [20,21].

In a cohort study conducted in the Netherlands [22], we investigated whether adolescents’ emotional recognition competency interacted with the family environment (e.g., parenting style) to affect depressive and anxiety disorders. This study assumed that adolescents’ emotional recognition could be specialized by distinguishing between positive and negative emotions and that they would be psychologically advantageous when placed in a family environment consistent with this emotional recognition specialization. The results of this study showed that adolescents specializing in positive emotional recognition had a low probability of developing anxiety disorders in an environment with parental emotional warmth but a relatively high probability of developing anxiety disorders when exposed to parental rejection. However, during the development of depression, children exposed to parental rejection had an increased risk of depression regardless of emotional recognition specialization. In the study, it was difficult to conclude that there is a direct relationship between parenting behavior and the probability of developing anxiety and depressive disorders in adolescence; it is emphasized that the type of emotional recognition specialization in adolescence can respond depending on the situation. In addition, exploring the meaning of emotional recognition interventions in adolescence is necessary.

The present study explores whether rejective parenting, internalized depression, and consequent externalized aggression in adolescents are regulated by emotional recognition abilities. This study aims to establish whether emotional recognition ability can mitigate the effects of negative parenting on adolescent depression and aggression and suggest a direction for emotional recognition competency strengthening interventions for adolescents with emotional and behavioral problems.

## 2. Method

### 2.1. Participants

The data used in the present study were extracted from the 4th year (survey in 2021) of the first-grade middle schoolers of the “Korea Children and Youth Panel Survey (KCYPS) 2018”, and the responses of 2265 first-grade high school students were considered. Of the 2265 students, 1172 were male (52%), and 1093 were female (48%). The Korean Children and Youth Panel Survey (KCYPS 2018), which started in 2018, was conducted to establish data that could identify the individual development and developmental environments of children and adolescents over time, targeting the same cohort in Korea in 2018. The study population comprised 3213 schools, 16,501 classes, and 448,816 students (first-grade middle schoolers) across 17 provinces. The sample was proportionally allocated based on the number of students in each city and province, the population was stratified, and the sample was extracted using the probability sampling method. The original panel extracted through this method consisted of 2590 people; in this study, the data of 2265 people who were maintained until the fourth year were included in the analysis. The survey method involved an individual interview survey using a tablet PC, and the survey contents included personal development and their development environments. All research procedures in this study were approved by the Institutional Review Board of Dankook University (approval number: 2023-08-001).

### 2.2. Measures 

#### 2.2.1. Rejective Parenting

The Parents as Social Context Questionnaire for Korean Adolescents (PSCQ_KA) used in the KCYPS was developed by Kim and Lee [23], and this study used four items of parents’ rejective parenting perceived by adolescents. The rejective parenting items were as follows: “Sometimes I wonder if my parents like me”, “My parents think I get in the way”, “My parents make me feel unnecessary”, and “My parents are not satisfied with anything I do”. In this case, a 4-point Likert scale was used; the higher the score, the higher the degree of rejective parenting attitude. In this case, Cronbach’s α was 0.805.

#### 2.2.2. Aggression

The aggression items used in the KCYPS were the six items of the self-report rating scale for children’s emotional and behavioral problems developed by Cho and Lim [24]. The scale consisted of six items: “There are times when I find fault with small things”, “There are times when I interfere with what others are doing”, “I argue or attack when I do not want what I want”, “I fight over trivial matters”, “There are times when I get angry all day long”, and “I cry for no reason”. On a 4-point Likert scale, the higher the score, the higher the degree of aggression. In this case, Cronbach’s α was 0.846.

#### 2.2.3. Depression

The depression measurement items used in the KCYPS were modified and used, except for 3 out of the 13 items developed as a simple mental diagnostic test by Kim et al. [25]. A total of 10 items, including “I am unhappy, sad, depressed”, “I have thoughts of wanting to die”, and “I am good at crying”, were used. In this case, Cronbach’s α was 0.904.

#### 2.2.4. Emotional Recognition Competency: Reading the Mind in the Eyes Test (RMET)

Emotional recognition competency was assessed using the Reading the Mind in the Eyes Test (RMET), which is the most widely used tool for assessing the ability to recognize the emotions of others. Baron-Cohen et al. [26] developed RMET to enable people to recognize and understand others’ emotions through their eyes and facial expressions. A total of 28 items were presented, and participants were asked to select an emotional state appropriate for the stimulus from four options. This metric is the sum of the answers answered correctly for 28 questions, and its value can range from 0 to 28 points (the higher the score, the higher the emotional recognition competency).

## 3. Statistical Analysis

To examine whether the direct path through which negative parenting attitudes affect aggression and the indirect path from negative parenting attitudes to depression and aggression vary according to individual emotional recognition abilities, Hayes [27] PROCESS macro was used. To verify the proposed hypothesis, the moderated mediation effect was tested using the PROCESS Model 59. Using this model, we tested the moderating effect of emotional recognition ability on both the direct and indirect pathways. The macro used bootstrap confidence intervals to estimate the conditional indirect relationships depending on the moderator variable, the effect of the independent variable on the dependent variable, the effect of the independent variable on the mediator variable, and the indirect effect of the independent variable through the mediator on the dependent variable.

## 4. Results

Table 1 presents the general demographic information of those in the sample. Table 2 presents the significant positive and negative correlations among the variables included in this study. Figure 1 illustrates the path of analysis used in this study.

To verify whether emotional recognition competency moderates the relationship between rejective parenting and aggression, moderated mediation modeling was estimated using the Hayes PROCESS macro model 59. As a result, the moderating effect of emotional recognition competency did not appear in the direct path in which rejective parenting influenced aggression.

Regarding the conditional indirect effects, the effect of rejective parenting on depression and the indirect effect of rejective parenting attitudes and depression on aggression varied according to individual emotional recognition abilities. Table 3 presents the suitability verification results of the model. To verify the statistical significance of the mediated moderating effect, bootstrapping was performed using the M and M ± 1 SD values of emotional recognition, and the results are presented in Table 4. As a result of our analysis, both the lower and upper limits of the indirect effect were statistically significant. Table 4 shows that the level of indirect effects varies according to emotional recognition ability level.

The higher the level of emotional recognition ability, the lower the depression affected by rejective parenting attitudes and aggression mediated by depression. The moderating effect of emotional recognition ability level on depression was low level of RMET (b = 1.061, SE = 0.076, 95% CI = 0.911–1.211), middle level of RMET (b = 0.831, SE = 0.047, 95% CI = 0.738–0.925), and high level of RMET (b = 0.729, SE = 0.063, 95% CI = 0.605–0.853). The moderation effect of emotional recognition ability level on aggression mediated by depression was low level of RMET (b = 0.395, SE = 0.019, 95% CI = 0.365 to 0.433), middle level of RMET (b = 0.328, SE = 0.011, 95% CI = 0.306–0.350), and high level of RMET (b = 0.298, SE = 0.014, 95% CI = 0.271 to 0.326). Thus, the proposed moderated mediating effect hypothesis was supported. In other words, the higher the emotional recognition competency level, the lower the degree of aggressive behavior through depression, according to negative parenting attitudes.

## 5. Discussion

In this study, adolescents’ emotional recognition competency was a significant moderating variable in the pathway of parental rejection of aggressive behavior mediated by depression in adolescence. However, the moderating effect of emotional recognition competency on the direct path from rejective parenting attitudes to aggression was not significant. We found that high emotional cognitive abilities did not directly lower aggressive behaviors caused by parental rejection but decreased aggressive behaviors preceded by a reduction in depression. One of the factors linking depression and aggression is isolation, and low actual or perceived social support can contribute to aggression [28]. Hale et al. [5] have already suggested that perceived parental rejection (mediated through adolescent depression) explains adolescent aggressive behavior, as tested by a mediation model. We recently revealed the path through which emotional recognition competency exerts a moderating effect and suggested a common factor that may influence depression and aggression

Emotional recognition competency is a transdiagnostic factor for early mental health problems, playing an important role in the maintenance and exacerbation of psychological problems [29]. A bias toward perceiving ambiguous emotions as sad or dissatisfied with aggressive intent can negatively change an individual’s emotional state. For example, negative biases in emotional processing are associated with the persistence of depression [30]. In addition, the lack of an ability to understand others’ feelings makes it difficult to suppress antisocial behaviors because one is unable to sympathize with and interpret others’ distress accurately, leading to negative and aggressive reactions from others [31].

A rejective parenting attitude Is considered to be one of the main antecedent factors causing emotional instability, depression, and aggressive behavioral problems. This is because the parent–child relationship (one’s first social relationship) affects the holistic growth and development of children in the areas of cognition, emotion, behavior, and socialization [32]. In previous studies, the more negative the parenting attitude perceived by a child, the higher the aggression [33].

One’s early family environment plays an important role in the development of emotional recognition. Children with emotional problems are more likely to grow up in disadvantaged and less supportive environments. Parents and children share some characteristics of emotional dysfunction, and families in emotionally deprived environments may exhibit similar deficits in emotional recognition [34]. Depending on the degree of deficit, emotional recognition competency can act as a risk factor for mental health problems [15,17], but children’s emotional recognition competency can develop as they grow up. A study on enhancing the emotional recognition of happiness among British university students reported that the modification of emotional recognition could increase positive emotional experiences [17]. In addition, in several studies of children with disruptive/aggressive behavioral disorders, content related to the promotion of emotional recognition has been included as part of intervention programs (e.g., PATHS (Promoting Alternative tHinking Strategies) [35,36]). In a study by Dadds et al. [15], as a result of trying to implement emotional recognition training intervention in various clinical groups of children (with conduct disorders, ADHD, autism, depression, and anxiety disorder), a significant improvement in emotional recognition was found in children with high-level conduct problems. In addition, a study by Wells et al. [37] on children with behavioral problems in problematic home environments commissioned by police community support officers, Cardiff emotional recognition training (CERT) focused on facial features, and facial expression identification was used. After six months of emotional recognition training, it was demonstrated that improved emotional recognition was associated with improvements in child behavior and mental well-being.

Adolescence is a period during which the structural development of the brain, reorganization of synapses, and functional maturation occur continuously and mentalization, the theory of mind, and emotional recognition continue to develop [38,39]. The development of adolescents’ social skills during this period can lead to clear changes in their individual psychosocial well-being and changes in their social relationships with their peers and family [40]. In addition to the existing results showing that developing emotional recognition capabilities is helpful for children with emotional problems, this study took environmental factors into account and included parents’ rejecting attitudes in the analysis. It was revealed that the path leading to depression and aggression from parents’ rejecting attitudes can be moderated depending on the adolescent’s individual emotional recognition developmental level. The present study has some limitations. Firstly, it proved difficult to clarify causal relationships due to this study being a cross-sectional study rather than a longitudinal study. Longitudinal data analysis is needed in future research. Secondly, the number of items regarding rejective parenting and aggression included in the panel survey was small, with four and six questions, respectively. More robust measures with more items need to be used in future investigations. Thirdly, the tools used to measure rejective parenting, aggression, and depression were self-reporting measures. Multi-method approaches such as clinical interviews, naturalistic observations, and experimental methods need to be applied. Because the RMET was also self-reporting in nature, it also needs to be performed in a laboratory to ensure appropriate participant performance and allow the participant to maintain their concentration during the test. 

However, revealing the moderating variables of the corresponding pathway is meaningful, as it suggests the need for intervention to improve emotional recognition competency at an exploratory level in dealing with the emotional and behavioral problems of children and adolescents caused by dysfunctional parenting. In addition, this study contributes to the generalization of the results in that a wide range of youths sampled regionally and equally by city size in Korea were the subjects of the study. Through the collection of national panel data, which will continue in the future, changes in emotional recognition competency can be observed using the Reading the Mind in the Eyes Test (RMET), which is considered to be a prototype of the theory of mind (the ability to reason about others’ emotions) for emotional states. If collected at a longitudinal level, we can expect to identify changes more clearly in individual depression and aggressive problems according to changes in the developmental level of emotional recognition competency. 

Additionally, in Korea, the impact of negative parenting attitudes, including rejective parenting, on children’s aggression and depression has been repeatedly demonstrated [41,42,43,44]. However, in Korean society, children recognize that there are significant differences in the roles of the father and mother, and the importance of the father and mother’s parenting attitudes appears to be distinct (e.g., in one study, regarding the father’s parenting attitude, the ‘authoritarian parenting attitude’ and ‘family values/discipline’ clusters were evaluated as having high importance, and in terms of the mother’s parenting attitude, the ‘dedication/overprotection’ and ‘achievement orientation/excessive expectations’ cluster were evaluated as having high importance) [45]. In future research, it would be helpful to distinguish the parenting styles of both parents when analyzing the relationship of parenting and the emotional problems of children.

It is also necessary to explore the changes that occur after providing emotional recognition competency strengthening interventions for children and adolescents with internal and external problems caused by problems in their parenting environment. In addition, education and intervention on the recognition of others’ and one’s own emotions are needed for all children and adolescents in educational settings, (i.e., schools), as an adolescent’s emotional recognition capacity is a domain that can be changed and improved. This is because the ability to reason about others’ mental and emotional states is closely related to the ability to introspect and rationalize one’s own mental and emotional states. If it is established that these attempts can enhance an individual’s emotional recognition capacity, individuals may more easily understand and recognize social cooperation signals from social members, including their peers and families, and better recognize and deal with individual emotional and behavioral problems.

## 6. Conclusions

The results of this study suggest that emotional recognition competency does not directly affect the relationship between negative parenting and aggression but significantly moderates the indirect pathway through depression. This suggests that the ability to accurately recognize and understand emotions (measured through Reading the Mind in the Eyes Test) can moderate the emotional and behavioral problems of adolescents affected by negative parenting. The significance of this study primarily lies in the importance of emotional recognition competency influencing adolescent mental health and behavior and secondarily lies within the possibility of establishing interventions focused on improving adolescent emotional recognition. By improving emotional recognition, adolescents can engage in social interaction, reduce conflict, and gain more support from their peers and family. This can lead to a decrease in depression and aggressive behavior. Schools and other educational settings may consider incorporating training or activities that promote emotional recognition competency into their education. Enhancing emotional recognition competency can help prevent or mitigate emotional and behavioral problems in adolescents.

## Figures and Tables

**Figure 1 ijerph-20-06775-f001:**
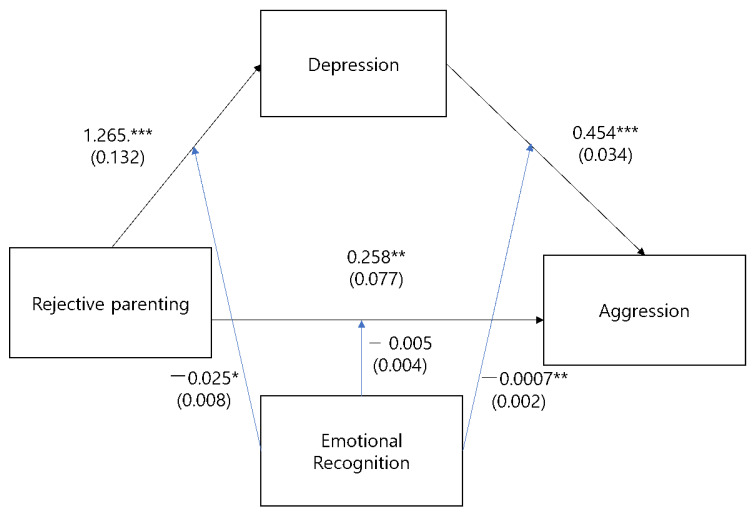
Coefficient of path model. Note. * *p* < 0.05; ** *p* < 0.01. *** *p* < 0.001.

**Table 1 ijerph-20-06775-t001:** Demographic information.

	N	%
**Total**	2265	
**Sex**		
Male	1172	52
Female	1093	48
**Father’s Education**		
high school graduate or less	613	27
college graduate	1365	60
graduate school or higher	184	8
do not know/no response	104	5
**Mother’s Education**		
high school graduate or less	603	27
college graduate	1463	65
graduate school or higher	137	6
do not know/no response	62	3
**Household income**		
Less than 1 million won	17	1
Less than 1 to 2 million won	73	3
Less than 2~3 million won	119	5
Less than 3~4 million won	323	14
Less than 4~5 million won	406	18
5 million won or more	1325	58
do not know/no response	3	0.1

**Table 2 ijerph-20-06775-t002:** Correlations between rejective parenting, depression, aggression, and RMET.

	Rejective Parenting	Depression	Aggression	RMET
Rejective parenting	*-*	0.343 **	0.345 **	−0.232 **
Depression		-	0.586 **	0.048 *
Aggression			-	−0.144 **
RMET				-

Note. * *p* < 0.05; ** *p* < 0.01.

**Table 3 ijerph-20-06775-t003:** Testing the moderated mediation effect of rejective parenting on aggression.

Predictors	Model 1 (Depression)		Model 2 (Aggression)	
	*b*	*t*	*B*	*t*
Rejective Parenting	1.265	9.532 ***	0.258	3.35 ***
Emotional Recognition	0.313	5.051 ***	0.108	2.783 **
Rejective Parenting x Emotional Recognition	−0.025	−3.206 **	−0.005	−1.116
Depression			0.454	13.274 ***
Depression x Emotional Recognition			−0.007	−3.738 ***
*F*	121.515 ***		351.928 ***	
*R-sq*	0.138		0.383	

Note. ** *p* < 0.01, *** *p* < 0.001.

**Table 4 ijerph-20-06775-t004:** Conditional indirect effect of emotional recognition.

**Rejective Parenting for Depression Moderated by Emotional Recognition**				
**Condition**	*B*	*SE*	Boot LLCI	Boot ULCI
**Low**	1.061	0.076	0.911	1.211
**Middle**	0.831	0.047	0.738	0.925
**High**	0.729	0.063	0.605	0.853
**Depression for Aggression Moderated by Emotional Recognition**				
**Condition**	** *B* **	** *SE* **	**Boot LLCI**	**Boot ULCI**
**Low**	0.395	0.019	0.356	0.433
**Middle**	0.328	0.011	0.306	0.350
**High**	0.298	0.014	0.271	0.326

## Data Availability

https://www.nypi.re.kr/archive, accessed on 14 September 2023.

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
