# Peer review of "Moderating Effects of Emotional Recognition Competency in Rejective Parenting and Adolescent Depression and Aggression"

_ijerph, 2023, doi:10.3390/ijerph20186775_

Round 1

Reviewer 1 Report

This cross-sectional analysis of survey data from the Korean Child and Youth Panel Survey from 2021 demonstrated that adolescents’ emotional recognition competency was a significant moderating variable in the pathway parental rejection of aggressive behavior mediated by depression in adolescents. There was no significant evidence of emotional recognition competency in moderating the direct pathway between parental rejection to aggressive behavior. As the authors indicated, these findings are of interest because emotional recognition competency appears to be modifiable in adolescents. The authors however needed to consider the following:

1. The measurement of parental rejection was limited to 4 items and aggression was limited to 3 items so the reliability and validity of these measures, beyond Cronbach’s alpha, were not addressed. The use of these measures should at the very least be mentioned as a limitation of the study and more robust measures should be used in future investigations.

2. In table 4, the low level of emotional recognition moderation bootstrap confidence interval include 1.0 and would appeared to have been a non-significant result. The authors should have addressed this in the results and discussion. 

3. In the discussion, the authors should have addressed that future studies and longitudinal research should adopt multi-method approaches to measure the concepts of parental rejection, adolescent aggression, adolescent depression and emotional recognition competency.

4. A few minor editorial corrections were needed to current manuscript.

5. On page 7, line 240, the authors needed to explain what was meant by “cardiac emotional recognition training”?

A few minor editorial corrections were needed to current manuscript.

Author Response

Reviewer 1

Comments

Authors’ Response

1.     The measurement of parental rejection was limited to 4 items and aggression was limited to 3 items so the reliability and validity of these measures, beyond Cronbach’s alpha, were not addressed. The use of these measures should at the very least be mentioned as a limitation of the study and more robust measures should be used in future investigations.

3.     In the discussion, the authors should have addressed that future studies and longitudinal research should adopt multi-method approaches to measure the concepts of parental rejection, adolescent aggression, adolescent depression and emotional recognition competency.

Based on comments 1 and 3, we have revised the content in the discussion section.

This study has some limitations. First, in that it is difficult to clarify the causal relationship because it is a cross-sectional study rather than a longitudinal study. Longitudinal data analysis is needed in future research. Second, the number of items about rejective parenting and aggression included in the panel survey was small, with 4 and 6 questions, respectively. More robust measures with more items need to be used in future investigation. Third, the tools to measure rejective parenting, aggression, and depression are self-report measures. Multi-method approaches such as clinical interview, naturalistic observation, and ex-perimental methods need to be applied. Because the RMET was also self-reported, it also needs to be performed in the laboratory to ensure the participant's appropriate performance and maintenance of concentration during the performance.

2. In table 4, the low level of emotional recognition moderation bootstrap confidence interval include 1.0 and would appeared to have been a non-significant result. The authors should have addressed this in the results and discussion. 

Bootstrapping was performed to verify the statistical significance of the mediated moderating effect, and it was considered statistically significant when both the lower and upper limits of the indirect effect did not include 0. We added the relevant phrase to the results and modified the result table 4.

To verify the statistical significance of the mediated moderating effect, bootstrapping was performed using the M and M±1SD values of the emotional recognition, and the results are presented in Table 4. As a result of the analysis, both the lower and upper limits of the indirect effect do not contain 0, which is statistically significant.

4. A few minor editorial corrections were needed to current manuscript.

The structure and expression of the sentences were overall revised. Also, conclusion section was added.

5. On page 7, line 240, the authors needed to explain what was meant by “cardiac emotional recognition training”?

There was an error in the wording, and it has been corrected.

Cardiff emotional recognition training (CERT)

Reviewer 2 Report

1.       On page 4, line#142-#150, authors will report the Cronbach's alpha value of RMET in this study.

2.       From bottom of page 6 bottom to top of page 7, line#211-#213, the comment “…, it was found that high emotional cognitive abilities….by a reduction in depression” will need authors’ comparison their findings with prior studies and provision of probable explanation of such findings as well as its consistency or contradiction.

3.       Since there was only one prior study on Korean children among all cited studies, authors will consider or explain how Korean culture on parenting style and children's behaviors may relate to the findings in this study.

Author Response

Reviewer 2

Comments

Authors’ Response

1.       On page 4, line#142-#150, authors will report the Cronbach's alpha value of RMET in this study.

RMET is a measure of ability (emotional recognition ability), and each question has a correct answer, so we did not calculate Cronbach's alpha.

2.       From bottom of page 6 bottom to top of page 7, line#211-#213, the comment “…, it was found that high emotional cognitive abilities….by a reduction in depression” will need authors’ comparison their findings with prior studies and provision of probable explanation of such findings as well as its consistency or contradiction.

In this study, adolescents’ emotional recognition competency was a significant moderating variable in the pathway of parental rejection of aggressive behavior mediated by depression in adolescence. However, the moderating effect of emotional recognition competency on the direct path from rejective parenting attitudes to aggression was not significant. We found that high emotional cognitive abilities did not directly lower aggressive behaviors caused by parental rejection but decreased aggressive behaviors preceded by a reduction in depression. One of the factors linking depression and aggression is isolation, and low actual or perceived social support can contribute to aggression [28]. Hale et al.[5] have already suggested that perceived parental rejection, mediated through adolescent depression, explains adolescent aggressive behavior as tested by a mediation model. We newly revealed the path through which emotional recognition competency moderating affect and suggest common factor that may have a influence on depression and aggression

Emotional recognition competency is a transdiagnostic factor for early mental health problems and plays an important role in the maintenance and exacerbation of psychological problems [29]. A bias toward perceiving ambiguous emotions as sad or dissatisfied with aggressive intent can negatively change an individual's emotional state. For example, negative biases in emotional processing are associated with the persistence of depression [30]. In addition, lacking ability to understand others’ feelings makes it difficult to suppress antisocial behaviors because one is unable to sympathize with and interpret others' pain accurately, thus leading to negative and aggressive reactions from others [31].

3.       Since there was only one prior study on Korean children among all cited studies, authors will consider or explain how Korean culture on parenting style and children's behaviors may relate to the findings in this study.

Additionally, in Korea, the impact of negative parenting attitudes including rejec-tive parenting, on children's aggression and depression has been repeatedly demonstrated [41-44]. However, in Korean society, children recognize that there are significant differences in the roles of father and mother, and the importance of the father and mother's parenting attitudes appears to be distinct(e.g., In the father’s parenting attitude, the ‘authoritarian parenting attitude’ and ‘family values/discipline’ clusters were evaluated as having high importance, and in the mother’s parenting attitude, the ‘dedication/overprotection’ and ‘achievement orientation/excessive expectations’ cluster were evaluated as having high importance)[45]. In future research, it would be helpful to distinguish the parenting styles of both parents when analyzing path of parenting and children’s emotional problems.